# High Sedentary Behaviour and Low Physical Activity are Associated with Anxiety and Depression in Myanmar and Vietnam

**DOI:** 10.3390/ijerph16071251

**Published:** 2019-04-08

**Authors:** Supa Pengpid, Karl Peltzer

**Affiliations:** 1ASEAN Institute for Health Development, Mahidol University, Salaya, Phutthamonthon, Nakhonpathom 73170, Thailand; supaprom@yahoo.com; 2Deputy Vice Chancellor Research and Innovation Office, North West University, Potchefstroom 2531, South Africa

**Keywords:** sedentary behaviour, physical activity, anxiety, depression, chronic diseases, adults, Myanmar, Vietnam

## Abstract

The study aimed to estimate independent and combined associations of sedentary behaviour and physical activity with anxiety and depression among chronic disease patients in Myanmar and Vietnam. The cross-sectional sample included 3201 chronic disease patients (median age 51 years, interquartile range 25) systematically recruited from primary care facilities in 2015. Sedentary time and physical activity were assessed with the General Physical Activity Questionnaire (GPAQ). Overall, the prevalence of sedentary time per day was 51.3% < 4 h, 31.2% between 4 and 8 h, and 17.5% 8 or more hours a day), and 30.7% engaged in low physical activity, 50.0% moderate, and 23.6% high physical activity. The prevalence of anxiety and depression was 12.7% and 19.9%, respectively. In the final logistic regression model, adjusted for relevant confounders, higher sedentary time (≥8 h) did not increase the odds for anxiety or depression, but moderate to high physical activity decreased the odds for anxiety and depression. Combined regression analysis found that participants with both less than eight hours of sedentary time and moderate or high physical activity had significantly lower odds of having anxiety and depression. Findings suggest an independent and combined association between moderate or high physical activity and low sedentary time with anxiety and/or depression among chronic disease patients in Myanmar and Vietnam.

## 1. Introduction

Anxiety and depression disorders contribute significantly to the “global burden of disease” [1,2].

“Anxiety is an emotion characterized by feelings of tension, worried thoughts and physical changes like increased blood pressure. People with anxiety disorders usually have recurring intrusive thoughts or concerns. They may avoid certain situations out of worry” [3]. “Depression causes feelings of sadness and/or a loss of interest in activities once enjoyed. It can lead to a variety of emotional and physical problems and can decrease a person’s ability to function at work and at home” [4]. High prevalence of anxiety disorder (17.0%) and depressive disorder (39.1%) were found in a sample of chronic disease patients in three Southeast Asian countries (Cambodia, Myanmar, and Vietnam) [5]. In the same population, 30.9% engaged in low physical activity [6], and 20.7% reported high sedentary behaviour (≥6 h/day) [7]. “Sedentary behavior refers to certain activities in a reclining, seated, or lying position requiring very low energy expenditure. It has been suggested to be distinct from physical inactivity and an independent predictor of metabolic risk even if an individual meets current physical activity guidelines” [8]. People with chronic conditions are more likely to have anxiety and depressive disorders than the general population [5], and are more likely to engage in low physical activity or sedentary behaviour than the general population [9].

Two recent systematic reviews found that sedentary behaviour increases the risk of anxiety [10] and depression [11]. Physical activity has benefits for physical health [12], and has also been shown to have benefits for the treatment of depression [13]. Interventions to reduce sedentary behaviour and increase physical activity may reduce anxiety [14] and depression [15]. Higher sedentary time and inadequate physical activity may have more deleterious effects on persons with metabolic risk factors and other chronic conditions [16].

The distinction between sedentary behaviour and physical activity in relation to anxiety and depression is not often investigated [12]. In a study among Chinese young adults, high sedentary time and low physical activity were associated with higher odds of anxiety and depression, both independent of each other and in combination [17]. In another study among Chinese young adults, low screen time and high physical activity decreased the odds of depression but not anxiety, and a combined inverse relationship was found for combined effects of low screen time and high physical activity on depression [18]. More research is needed to investigate the independent and combined effects of sedentary behaviour and physical activity on mental health, in particular in vulnerable populations, such as those with chronic diseases.

This study aimed to assess the independent and combined associations of sedentary behaviour and physical activity with anxiety and depression among chronic disease patients in Myanmar and Vietnam.

## 2. Materials and Methods

### 2.1. Sample and Procedure

In Myanmar and Vietnam, a cross-sectional survey in rural and urban primary health facilities was conducted with out-patients 18 years and older and with chronic diseases (more details can be found in Reference [18]). Patients provided informed consent prior to an interview-administered questionnaire. The study protocol was approved in Myanmar by the Research and Ethical Committee, University of Medicine 1, Yangon; in Vietnam by the Committee for Research Ethics, Hanoi School of Public Health; and in Thailand by the Committee for Research Ethics in Social Sciences, Mahidol University (COA. No.: 2014/193.0807) [19].

### 2.2. Measures

#### 2.2.1. Exposure Variables

Sedentary behaviour was assessed with the General Physical Activity Questionnaire (GPAQ) [20]. Starting with the explanatory statement: “The following question is about sitting or reclining at work, at home, getting to and from places, or with friends, including time spent (sitting at a desk, sitting with friends, travelling in car, bus, train, reading, playing cards or watching television), but do not include time spent sleeping”, and the question was: “How much time do you usually spend sitting or reclining on a typical day?” [20]. Sedentary time was categorized into less than 4 h, between 4 and 8 h, or 8 or more hours a day [21].

Physical activity was assessed with the GPAQ [20,22]. Using GPAQ guidelines [20], results were classified into low, moderate, and high physical activity. “High physical activity: A person reaching any of the following criteria is classified in this category: Vigorous-intensity activity on at least 3 days achieving a minimum of at least 1500 MET (metabolic equivalent)-minutes per week or; 7 or more days of any combination of walking, moderate or vigorous intensity activities achieving a minimum of at least 3000 MET-minutes per week. Moderate physical activity: A person not meeting the criteria for the ‘high’ category, but meeting any of the following criteria is classified in this category: 3 or more days of vigorous-intensity activity of at least 20 min per day or; 5 or more days of moderate-intensity activity or walking of at least 30 min per day or; 5 or more days of any combination of walking, moderate or vigorous intensity activities achieving a minimum of at least 600 MET-minutes per week. Low physical activity: A person not meeting any of the above mentioned criteria falls in this category” [22].

#### 2.2.2. Outcome Variables

Anxiety and depression were assessed with the Hospital Anxiety and Depression Scale (HADS) [23]. Participants scoring 11 or more on the HADS were classified as having moderate to severe anxiety and depression [23]. The HADS subscales had satisfactory reliability (α anxiety: 0.79; α depression: 0.70). The HADS is a “simple yet reliable tool for use in medical, including primary care, practice” [24].

#### 2.2.3. Confounding Variables

Confounding variables based on literature review [12,17,18,25,26,27,28] included sociodemographic variables, health behaviours and chronic conditions. Sociodemographic variables included age, sex, country, formal education and residential status [19]. Tobacco use was assessed with two questions: (1) “Do you currently use one or more of the following tobacco products (cigarettes, snuff, chewing tobacco, cigars, etc.)?” and (2) “Do you currently use any smokeless tobacco, such as snuff, chewing tobacco, betel?” Response options were “yes” or “no” [29]. Current tobacco use was defined as any tobacco use.

Problem drinking was assessed by the “Alcohol Use Disorder Identification Test (AUDIT)-C” [30]. Cronbach α for the AUDIT-C was 0.82 in this study. Fruit and vegetable consumption was measured with the question, “How many servings of fruit or vegetables do you eat per day” [31]. Responses were classified into: 0 = 0–2 and 1 = 3 or more.

Chronic conditions were assessed by self-reported health care provided diagnosed chronic conditions treated in the past 12 months for any of 21 chronic conditions, such as “asthma, chronic obstructive pulmonary disease (COPD), diabetes mellitus, hypertension, dyslipidemia, coronary artery disease, cardiac failure, cardiac arrhythmias, stroke, arthritis, cancer, gout and other musculoskeletal conditions, such as chronic backache, Parkinson’s disease, liver disease, kidney disease, thyroid disease, stomach and intestinal diseases, epilepsy and mental disorders” [19].

### 2.3. Data Analysis

The data were analysed with IBM SPSS Statistics for Windows (Version 25.0. Armonk, NY, USA: IBM Corp.). Parametric and non-parametric tests were used to calculate differences in proportions and medians. Multivariable logistic regression was used to calculate odds ratios (confidence intervals) of the independent associations of sedentary behaviour and physical activity with the prevalence of anxiety and depression. Model 1: Unadjusted, Model 2: Adjusted for sociodemographic, lifestyle factors and chronic conditions, and Model 3: Adjusted for Model 2 and sedentary behaviour or physical activity. We tested interactions between predictors and sex on anxiety and depression, but could not find any. Moreover, we used multivariable logistic regression to estimate the combined relationship between sedentary behaviour and physical activity with prevalence of anxiety and depression. Multivariable models were adjusted for relevant confounders, including age, sex, education, residence, tobacco use, problem drinking, fruit and vegetable consumption and number of chronic conditions. For the combined regression analysis, the sample was sub-divided based on sedentary and physical activity levels into four groups: (1) high sedentary time (≥8 h) plus low physical activity group (reference category); (2) high sedentary time (≥8 h) plus moderate or high physical activity group; (3) low or moderate sedentary time (<8 h) plus low physical activity group; and (4) low or moderate sedentary time (<8 h) plus moderate or high physical activity group. A *p*-value of <0.05 was considered significant.

## 3. Results

### 3.1. Sample Characteristics

The total sample included 3201 adults, 1600 from Myanmar and 1601 from Vietnam. The median age was 51 years (interquartile range = 25 years), and 65.1% were female. Overall, the study population engaged in less than 4 h (51.3%), between 4 and 8 h (31.2%), and 8 or more hours of sedentary time a day (17.5%). The physical activity levels of the respondents were 26.4% low, 50.0% moderate and 23.6% high physical activity. The prevalence of anxiety was 12.7%, and the prevalence of depression was 19.3% (see Table 1).

### 3.2. Associations between Sedentary Behaviour, Physical Activity and Anxiety and Depression

In the second logistic regression model, adjusted for sociodemographic, lifestyle factors and chronic conditions, higher sedentary time (≥8 h) increased the odds for anxiety and depression, and moderate to high physical activity decreased the odds for anxiety and depression. In the third logistic regression model, adjusted for sociodemographic, lifestyle factors, chronic conditions, and sedentary behaviour or physical activity, higher sedentary time (≥8 h) no longer increased the odds for anxiety and depression, but moderate to high physical activity decreased the odds for anxiety and for depression. Considering a greater *p*-value (*p* < 0.01) to minimize Type 1 error in model 2, adjusted for sociodemographic, lifestyle factors and chronic conditions, higher sedentary time (≥8 h) was no longer significantly associated with anxiety and depression (see Table 2).

### 3.3. Combined Effects of Sedentary Behaviour and Physical Activity on Anxiety and Depression

By dividing study participants into four sub-groups based on the levels of sedentary time and physical activity, we found that those reporting less than eight hours of sedentary time and engaging in moderate or high physical activity had significantly lower odds of having anxiety and depression, compared to participants with high sedentary time (≥8 h) and low physical activity, after adjusting for age, sex, education, residence, tobacco use, problem drinking, fruit and vegetable consumption and number of chronic conditions. Most of the effect sizes were of moderate magnitude (or <0.54) [32] (see Table 3).

## 4. Discussion

This study aimed to assess independent and combined associations of sedentary behaviour and physical activity with anxiety and depression among chronic disease patients in Myanmar and Vietnam. Consistent with previous reviews in the general population [3,4,7,8], this study found that higher sedentary time (≥8 h) increased the odds for anxiety and depression in the model without adjusting for physical activity, and moderate to high physical activity decreased the odds for anxiety and depression (after adjusting for relevant confounders), in this population of chronic disease patients. The effects of physical activity were stronger than those of sedentary behaviour on anxiety and depression. Similar results were found in a study among Japanese adults [26]. This may be confirmed from some of the reviews showing that sedentary behaviour has a small positive association with anxiety [10].

Finally, the study confirms previous evidence with university students in China [10,11] suggesting that there is a combined effect between physical activity and sedentary behaviour on anxiety and depression among chronic disease patients in Southeast Asia. The effects of combined physical activity and sedentary behaviour were stronger on depression than on anxiety. The dose-response relationship between physical activity and sedentary behaviour on anxiety and depression needs further research [33]. Dunn et al. suggest that “it is highly likely that depression, like high blood pressure, has multiple etiologies, and that exercise, acting on multiple biological and psychological systems, could lead to synergistic adaptations that effectively reduce symptoms of anxiety and depressive disorders” [33].

### Study Limitations

This investigation was cross-sectional in nature, and no causal conclusions can be drawn. Further, the exposure variables, sedentary behaviour and physical activity, were assessed by self-report and future studies should include objective measures as well. The assessment of the outcome variables (i.e., anxiety and depression) used only screening measures, and future studies may include diagnostic interviews. Variables such as body mass index were not assessed in this investigation and should be included in future studies.

## 5. Conclusions

Our findings not only add to the current results demonstrating a beneficial independent effect of low sedentary time and moderate or high physical activity on anxiety and/or depression in the general population, but also extending findings from a few studies showing combined effects of low sedentary time and moderate or high physical activity on anxiety and depression in chronic disease patients. Mental health promotion strategies among chronic disease patients may include reducing sedentary behaviour and increasing physical activity.

## Figures and Tables

**Table 1 ijerph-16-01251-t001:** Basic sample characteristics.

	Total (*n* = 3201)	Myanmar	Vietnam	Urban	Rural	Male (*n* = 1115)	Female (*n* = 2084)	*p*-Value
Age in years, Median (Interquartile range)	51 (25)	56 (21)	47 (33)	52 (27)	53 (26)	52 (25.5)	51 (25)	0.096
Country:								
Myanmar	1600 (50.0)		-	50.0	50.0	44.2	53.9	<0.001
Vietnam	1601 (50.0)			47.5	52.5	55.8	46.1	
Education:								
Grade 0–5	620 (19.4)	31.4	7.3	16.7	22.0	14.0	22.4	<0.001
Grade 6–11	1635 (51.2)	56.3	46.1	46.6	55.5	50.0	51.9	
Grade 12 or more	941 (29.4)	12.3	46.6	36.7	22.6	35.9	26.7	
Residence:								
Rural	1640 (51.3)	-	-	-	-	49.5	52.2	0.152
Urban	1561 (48.8)					50.5	47.8	
Current tobacco use	761 (24.6)	31.9	16.8	29.2	20.3	46.0	13.5	<0.001
Problem drinking	387 (12.2)	2.8	21.9	11.2	13.3	28.8	3.5	<0.001
Fruit and vegetable consumption	1811 (56.6)	51.6	61.5	59.8	53.5	54.5	58.1	0.050
Chronic conditions:								
One	1987 (62.1)	64.4	59.3	56.4	67.1	65.8	60.0	<0.001
Two	838 (26.2)	26	25.7	30.1	22.7	25.3	26.9	
Three or more	376 (11.7)	8.6	15.0	13.6	10.1	8.8	13.1	
Sedentary behaviour:								
<4 h	1643 (51.3)	40.4	62.2	60.4	42.6	52.8	50.1	0.019
4–8 h	998 (31.2)	38.4	24.0	29.2	33.1	32.2	30.8	
≥8 h	560 (17.5)	21.2	13.8	10.4	24.3	15.0	19.1	
Physical activity:								
Low	844 (26.4)	31.5	21.2	26.7	26.1	30.7	23.9	<0.001
Moderate	1602 (50.0)	52.9	47.2	47.2	52.7	37.3	56.7	
High	755 (23.6)	15.6	31.6	26.1	21.2	31.9	19.3	
Anxiety	401 (12.7)	10.3	15.3	12.1	13.3	14.5	11.8	0.031
Depression	629 (19.9)	25.5	14.2	19.9	20	19.3	20.4	0.482

**Table 2 ijerph-16-01251-t002:** Odds ratios for anxiety and depression according to levels of sedentary behaviour and physical activity.

Variable	Anxiety		Depression	
Model 1	UOR (95% CI)	*p*-value	UOR (95% CI)	*p*-value
Sedentary time a day				
<4 h	1 (Reference)		1 (Reference)	
4–8 h	0.99 (0.76, 1.26)	0.928	0.99 (0.81, 1.21)	0.937
≥8 h	1.42 (1.08, 1.86)	0.012	1.40 (1.11, 1.76)	0.004
Model 2	AOR ^1^ (95% CI)		AOR ^1^ (95% CI)	
Sedentary time a day:				
<4 h	1 (Reference)		1 (Reference)	
4–8 h	0.97 (0.75; 1.26)	0.817	0.94 (0.76; 1.17)	0.590
≥8 h	1.34 (1.01; 1.80)	0.048	1.30 (1.01; 1.69)	0.046
Model 3	AOR ^2^ (95% CI)		AOR ^2^ (95% CI)	
Sedentary time a day:				
<4 h	1 (Reference)		1 (Reference)	
4–8 h	0.93 (0.72; 1.21)	0.584	0.92 (0.74; 1.15)	0.457
≥8 h	1.17 (0.86; 1.58)	0.316	1.19 (0.91; 1.56)	0.194
Model 1	UOR (95% CI)		UOR (95% CI)	
Physical activity:				
Low	1 (Reference)		1 (Reference)	
Moderate	0.64 (0.50; 0.81)	<0.001	0.52 (0.43; 0.64)	< 0.001
High	0.58 (0.43; 0.78)	<0.001	0.46 (0.30; 0.58)	< 0.001
Model 2	AOR ^1^ (95% CI)		AOR ^1^ (95% CI)	
Physical activity				
Low	1 (Reference)		1 (Reference)	
Moderate	0.67 (0.52; 0.87)	<0.002	0.56 (0.45; 0.69)	<0.001
High	0.58 (0.43; 0.79)	<0.001	0.55 (0.41; 0.72)	<0.001
Model 3	AOR ^3^ (95% CI)		AOR ^3^ (95% CI)	
Physical activity				
Low	1 (Reference)		1 (Reference)	
Moderate	0.69 (0.54; 0.90)	0.006	0.57 (0.45; 0.71)	<0.001
High	0.61 (0.44; 0.84)	0.002	0.56 (0.42; 0.74)	<0.001

UOR: unadjusted odds ratio; AOR: adjusted odds ratio; CI: confidence interval; ^1^ Adjusted for age, sex, education, residence, tobacco use, problem drinking, fruit and vegetable consumption and number of chronic conditions; ^2^ Adjusted for model 2 and physical activity; ^3^ Adjusted for model 2 and sedentary behaviour.

**Table 3 ijerph-16-01251-t003:** Combined effects of sedentary behaviour and physical activity on anxiety and depression.

Variable	*n* (subgroup)	Anxiety		
		Yes, *n* (%)	UOR	AOR ^1,2^
High sedentary and low physical activity	244	53 (21.7)	1 (Reference)	1 (Reference)
High sedentary and moderate or high physical activity	311	37 (11.9)	0.49 (0.31; 0.77) **	0.58 (0.36; 0.94) *
Low or moderate sedentary and low physical activity	577	86 (14.9)	0.63 (0.43, 0.92) *	0.74 (0.89, 1.10)
Low or moderate sedentary and moderate or high physical activity	2019	225 (11.1)	0.45 (0.32, 0.63) ***	0.50 (0.35, 0.72) ***
		Yes, *n* (%)	UOR	AOR ^1,3^
		Depression		
High sedentary and low physical activity	246	80 (32.5)	1 (Reference)	1 (Reference)
High sedentary and moderate or high physical activity	309	57 (18.4)	0.47 (0.32, 0.70) ***	0.56 (0.37, 0.86) **
Low or moderate sedentary and low physical activity	582	158 (27.1)	0.77 (0.56, 1.06)	0.92 (0.64, 1.32)
Low or moderate sedentary and moderate or high physical activity	2020	334 (16.5)	0.41 (0.31, 0.55) ***	0.49 (0.36, 0.68) ***

UOR: unadjusted odds ratio; AOR: adjusted odds ratio; CI: confidence interval; ^1^ Adjusted for age, sex, education, residence, tobacco use, problem drinking, fruit and vegetable consumption and number of chronic conditions; *** *p* < 0.001; ** *p* < 0.01; * *p* < 0.05; ^2^ Nagelkerke R^2^ = 0.023; X^2^ = 4.93; *p* = 0.765; ^3^ Nagelkerke R^2^ = 0.13; X^2^ = 14.75; *p* = 0.023.

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
