# Peer review of "High Sedentary Behaviour and Low Physical Activity are Associated with Anxiety and Depression in Myanmar and Vietnam"

_ijerph, 2019, doi:10.3390/ijerph16071251_

Round 1
Reviewer 1 Report
1) Please review entire paper for spelling errors.
2) Introduction is well written and organized, including relevant and up to date background studies that effectively demonstrate the need for this investigation.
- Perhaps in the introduction more emphasis could be placed on the important distinction between sedentary behaviors and physical activity?
- Please review lines 19 and 20. this sentence is awkwardly written.
3) This study draws a critical distinction between sedentary behaviors and physical activity, and is especially impactful, tying these behaviors to anxiety and depression. Again, this a major strength that should be highlighted in the introduction.
- While this utilizes total number of chronic conditions as a possible co-variate, it would very valuable to see how presence of specific chronic conditions might impact these associations.
4) There are minor errors and inconsistencies throughout the paper. For example in table 2, at the bottom, it says "Odds Ration", instead of Odds Ratio. This is minor but needs to be changed. Please review the entire paper carefully.

Author Response
1) Please review entire paper for spelling errors.
R: Corrected
2) Introduction is well written and organized, including relevant and up to date background studies that effectively demonstrate the need for this investigation.
- Perhaps in the introduction more emphasis could be placed on the important distinction between sedentary behaviors and physical activity?
R: added
- Please review lines 19 and 20. this sentence is awkwardly written.
R: Corrected
3) This study draws a critical distinction between sedentary behaviors and physical activity, and is especially impactful, tying these behaviors to anxiety and depression. Again, this a major strength that should be highlighted in the introduction.
R: More is added
- While this utilizes total number of chronic conditions as a possible co-variate, it would very valuable to see how presence of specific chronic conditions might impact these associations.
R: there are too many chronic conditions to make this meaningful
4) There are minor errors and inconsistencies throughout the paper. For example in table 2, at the bottom, it says "Odds Ration", instead of Odds Ratio. This is minor but needs to be changed. Please review the entire paper carefully.
R: Corrected
Reviewer 2 Report
COMMENTS TO THE AUTHORS
The results presented in this manuscript are not novel, however, it is a good effort to assess these associations in the context of Asian countries as Myanmar and Vietnam, and it will be very useful to have some kind of comparisons between these countries to make this study more interesting.
In addition, there are some methodological and format concerns that must be attended, some of them are detailed below.
Abstract
Include in the abstract the period of the study.
Introduction
Include a brief definition of anxiety and depression and why it is important to make a distinction between them. In addition, more information about the magnitude of these conditions in Myanmar and Vietnam is needed, as well as, sedentary time and physical activity or inactivity data.
The introduction should state the rationale of approaches that are combining physical activity and sedentary, why are important and why the evidence about this research area has not been enough to understand these behaviors. In the introduction, there isn’t information that allows explaining why it is important to carry out this study, in comparison to other which have been carried out previously.
Line 48 “hugher” instead “higher”.
The aim should include basic information about the population in which this study was conducted.
Methods
The study as the authors stated was conducted among people with chronic conditions, however, these were not considered in the approach of the study and how these conditions could be related to sedentary time and physical activity. For the other hand, it is important to note that some kind of correlation with mental issues as depression and anxiety occurs with chronic conditions, as well.
In addition, it is important to know why this study was conducted among subjects with chronic conditions and the implications of this study population with chronic conditions both in the aim and analysis. i.e., an analysis stratified by chronic conditions or adjusted by chronic conditions at least will be required.
Include cut-off points of physical activity, why 3 levels.
As the authors did not include information enough information about the sample, it is important to know why they used the “Hospital Anxiety and Depression Scale” to assess depression and anxiety instead other scales, because the sample comes from primary health care settings.
It is important to consider some variables as potential confounders o modifiers effect because the relationship between physical activity, sedentary and mental issues could be observed in bidirectional ways thus it is important to have an adjust by other variables which were not included in the analyses.
As the sample was collected in rural and urban primary care facilities, some differences in sedentary time and physical activity could be expected, due to these possible variations, authors should consider some strategy of analysis to deal with these variations.
Line 88, the sentence is incomplete.
Results
I suggest editions in Table 1, including a description by country, not only as row but instead as a column. In that way, the study characteristics could also be described by country and for a rural and urban setting, as well.
Discussion
Line 141.... the study included only sociodemographic variables as adjust variables, and some of them are not relevant confounders, please reconsidered this statement.
In general, discussion needs to be a focus on understanding the associations of this study among sedentary time and physical activity with anxiety and depression in individual with chronic diseases. Authors need to rethinking discussion to have a wider approach to understand differences found it in the study.
Conclusions are wide general some specific statements related with this study are required, i.e. we don´t know what kind of chronic conditions were assessed and thus the conclusions are limited.
Authors contributions and funding information needs to be completed.
Author Response
Abstract
Include in the abstract the period of the study.
R: added
Introduction
Include a brief definition of anxiety and depression and why it is important to make a distinction between them. In addition, more information about the magnitude of these conditions in Myanmar and Vietnam is needed, as well as, sedentary time and physical activity or inactivity data.
R: added
The introduction should state the rationale of approaches that are combining physical activity and sedentary, why are important and why the evidence about this research area has not been enough to understand these behaviors. In the introduction, there isn’t information that allows explaining why it is important to carry out this study, in comparison to other which have been carried out previously.
R: added
Line 48 “hugher” instead “higher”.
R: Corrected
The aim should include basic information about the population in which this study was conducted.
R: Added
Methods
The study as the authors stated was conducted among people with chronic conditions, however, these were not considered in the approach of the study and how these conditions could be related to sedentary time and physical activity. For the other hand, it is important to note that some kind of correlation with mental issues as depression and anxiety occurs with chronic conditions, as well.
R: added
In addition, it is important to know why this study was conducted among subjects with chronic conditions and the implications of this study population with chronic conditions both in the aim and analysis. i.e., an analysis stratified by chronic conditions or adjusted by chronic conditions at least will be required.
R: the models are adjusted by the number of chronic conditions
Include cut-off points of physical activity, why 3 levels.
R: added
As the authors did not include information enough information about the sample, it is important to know why they used the “Hospital Anxiety and Depression Scale” to assess depression and anxiety instead other scales, because the sample comes from primary health care settings.
R: added
It is important to consider some variables as potential confounders o modifiers effect because the relationship between physical activity, sedentary and mental issues could be observed in
bidirectional ways thus it is important to have an adjust by other variables which were not included in the analyses.
R: added
As the sample was collected in rural and urban primary care facilities, some differences in sedentary time and physical activity could be expected, due to these possible variations, authors should consider some strategy of analysis to deal with these variations.
R: rural-urban is adjusted for in the analysis
Line 88, the sentence is incomplete.
R: Corrected
Results
I suggest editions in Table 1, including a description by country, not only as row but instead as a column. In that way, the study characteristics could also be described by country and for a rural and urban setting, as well.
R: added
Discussion
Line 141.... the study included only sociodemographic variables as adjust variables, and some of them are not relevant confounders, please reconsidered this statement.
R: This is not correct, it also included lifestyle factors and number of chronic conditions, the relevance has now been supported in the measures section
In general, discussion needs to be a focus on understanding the associations of this study among sedentary time and physical activity with anxiety and depression in individual with chronic diseases. Authors need to rethinking discussion to have a wider approach to understand differences found it in the study.
R: More is added, yet this is a brief report
Conclusions are wide general some specific statements related with this study are required, i.e. we don´t know what kind of chronic conditions were assessed and thus the conclusions are limited.
R: Changed
Authors contributions and funding information needs to be completed.
R: added
Reviewer 3 Report
This study examined independent and combined associations of physical activity and sedentary behaviour among adults in Myanmar and Vietnam. This study potentially provides important information on the research area in under-represented population groups. In addition, most previous literature predominantly examined physical health outcomes rather than mental health outcomes; thus, I believe that this study has some merits to the body of evidence. That being said, introduction and discussion are largely lacking in providing important background information such as contexts, rationale, and discussion is rather short and there is not really any in-depth interpretation along with important references that can be included. I believe that strengthening introduction and discussion, and providing more detailed in the methods section will improve the manuscript.
Introduction: Introduction should clearly lay out the study rationale/justification on why it is important to test this relationship in a particular population that is included in this study. Why looking at the relationship between lifestyle behaviours and mental health is important among adults in Myanmar and Vietnam? What is known with regard to the research questions in this particular population group? What is the study context? What this study adds to the literature that the authors included in the introduction?
Methods:
Lines 70-71: I understand that the appropriate citation is available in the section but please consider describing how PA scores were calculated in detail to make it easier for readers.
Lines 77-88: Why all these variables were included as confounders? any supporting documents suggesting the confounding effect of these variables on mental health and therefore should be included in this study?
To examine the "independent" effect, analysis involving sedentary behaviour and mental health should be controlled for physical activity and the same thing for PA. To examine the "interactive" effect, the manuscript should be testing some kind of interactions between two variables which can be argued that it did. However, I am not sure if "interactive" is the right word. Please also see my comment below (results 3)
Results:
1. Table 1 provides sample characteristics by sex and some test statistics indicate that there is a significant sex difference. Given that Sedentary behaviour, physical activity, and anxiety significantly differ by sex, I am wondering if the authors have considered to test interactions between predictors and sex on anxiety. If there is a significant interaction effect, the subsequent analyses should be stratified by sex.
2. Given the large sample size, a p-value for significance should be stricter to minimize Type 1 error for analyses that doesn't involve subgroups (Table 2 results).
3. Line 122: perhaps combined effects make more sense than interactive effects; interactive effects can be misleading
4. Table 3: The results for both tests are all borderline significant. Please interpret effect size given that the observed associations are actually trivial.
Discussion: Discussion largely lacks details and missing in-depth of the interpretation of the results against/along with previous literature. Also, the formatting for discussion for an academic journal should be 4-5 paragraphs discussing about the main results, implications, and study strengths as well as limitations. There is accumulating evidence suggesting the combined effect of PA and SB on mental health and those studies (especially reviews) should be included in the overall discussion.
Author Response
Methods:
Lines 70-71: I understand that the appropriate citation is available in the section but please consider describing how PA scores were calculated in detail to make it easier for readers.
R: added
Lines 77-88: Why all these variables were included as confounders? any supporting documents suggesting the confounding effect of these variables on mental health and therefore should be included in this study?
R: support for confounders is added
To examine the "independent" effect, analysis involving sedentary behaviour and mental health should be controlled for physical activity and the same thing for PA. To examine the "interactive" effect, the manuscript should be testing some kind of interactions between two variables which can be argued that it did. However, I am not sure if "interactive" is the right word. Please also see my comment below (results 3)
R: added
Methods:
Lines 70-71: I understand that the appropriate citation is available in the section but please consider describing how PA scores were calculated in detail to make it easier for readers.
R: added
Lines 77-88: Why all these variables were included as confounders? any supporting documents suggesting the confounding effect of these variables on mental health and therefore should be included in this study?
R: support for confounders is added
To examine the "independent" effect, analysis involving sedentary behaviour and mental health should be controlled for physical activity and the same thing for PA. To examine the "interactive" effect, the manuscript should be testing some kind of interactions between two variables which can be argued that it did. However, I am not sure if "interactive" is the right word. Please also see my comment below (results 3)
R: added
Results:
1. Table 1 provides sample characteristics by sex and some test statistics indicate that there is a significant sex difference. Given that Sedentary behaviour, physical activity, and anxiety significantly differ by sex, I am wondering if the authors have considered to test interactions between predictors and sex on anxiety. If there is a significant interaction effect, the subsequent analyses should be stratified by sex.
2. R: added
3. Given the large sample size, a p-value for significance should be stricter to minimize Type 1 error for analyses that doesn't involve subgroups (Table 2 results).
R: added
3. Line 122: perhaps combined effects make more sense than interactive effects; interactive effects can be misleading
R: changed accordingly
4. Table 3: The results for both tests are all borderline significant. Please interpret effect size given that the observed associations are actually trivial.
R: This is redone, and effect sizes added
Discussion: Discussion largely lacks details and missing in-depth of the interpretation of the results against/along with previous literature. Also, the formatting for discussion for an academic journal should be 4-5 paragraphs discussing about the main results, implications, and study strengths as well as limitations.
R: More is added, yet this is a brief report
There is accumulating evidence suggesting the combined effect of PA and SB on mental health and those studies (especially reviews) should be included in the overall discussion.
R: which are those, there are no reviews on combined effect…, the individual studies are included
Results:
1. Table 1 provides sample characteristics by sex and some test statistics indicate that there is a significant sex difference. Given that Sedentary behaviour, physical activity, and anxiety significantly differ by sex, I am wondering if the authors have considered to test interactions between predictors and sex on anxiety. If there is a significant interaction effect, the subsequent analyses should be stratified by sex.
2. R: added
3. Given the large sample size, a p-value for significance should be stricter to minimize Type 1 error for analyses that doesn't involve subgroups (Table 2 results).
R: added
3. Line 122: perhaps combined effects make more sense than interactive effects; interactive effects can be misleading
R: changed accordingly
4. Table 3: The results for both tests are all borderline significant. Please interpret effect size given that the observed associations are actually trivial.
R: This is redone, and effect sizes added
Discussion: Discussion largely lacks details and missing in-depth of the interpretation of the results against/along with previous literature. Also, the formatting for discussion for an academic journal should be 4-5 paragraphs discussing about the main results, implications, and study strengths as well as limitations.
R: More is added, yet this is a brief report
There is accumulating evidence suggesting the combined effect of PA and SB on mental health and those studies (especially reviews) should be included in the overall discussion.
R: which are those, there are no reviews on combined effect…, the individual studies are included
Round 2
Reviewer 2 Report
I haven't found a novel approach and in addition, suggestion about comparison among countries, it wasn't considered it. In the same way, some kind of analysis of chronic conditions would be interesting.
Author Response
I haven't found a novel approach and
R: We do not know of any other study investigating combined effects on anxiety/depression in chronic disease patients
in addition, suggestion about comparison among countries, it wasn't considered it.
R: The comparison among countries is in Table 1, as below
Table 1. Basic sample characteristics.
Total (n=3201) Myanmar Vietnam Urban Rural Male (=1115) Female (n=2084) P-value
Age in years, Median (Interquartile range) 51 (25) 56 (21) 47 (33) 52 (27) 53 (26) 52 (25.5) 51 (25) 0.096
Country
Myanmar
Vietnam
1600 (50.0)
1601 (50.0)
---
---
50.0
47.5
50.0
52.5
44.2
55.8
53.9
46.1
<0.001
Education
Grade 0-5
Grade 6-11
Grade 12 or more
620 (19.4)
1635 (51.2)
941 (29.4)
31.4
56.3
12.3
7.3
46.1
46.6
16.7
46.6
36.7
22.0
55.5
22.6
14.0
50.0
35.9
22.4
51.9
26.7
<0.001
Residence
Rural
Urban
1640 (51.3)
1561 (48.8)
---
---
---
---
49.5
50.5
52.2
47.8
0.152
Current tobacco use 761 (24.6) 31.9 16.8 29.2 20.3 46.0 13.5 <0.001
Problem drinking 387 (12.2) 2.8 21.9 11.2 13.3 28.8 3.5 <0.001
Fruit and vegetable consumption 1811 (56.6) 51.6 61.5 59.8 53.5 54.5 58.1 0.050
Chronic conditions
One
Two
Three or more
1987 (62.1)
838 (26.2)
376 (11.7)
64.4
26.0
8.6
59.3
25.7
15.0
56.4
30.1
13.6
67.1
22.7
10.1
65.8
25.3
8.8
60.0
26.9
13.1
<0.001
Sedentary behaviour
<4 hours
4- <8 hours
≥8 hours
1643 (51.3)
998 (31.2)
560 (17.5)
40.4
38.4
21.2
62.2
24.0
13.8
60.4
29.2
10.4
42.6
33.1
24.3
52.8
32.2
15.0
50.1
30.8
19.1
0.019
Physical activity
Low
Moderate
High
844 (26.4)
1602 (50.0)
755 (23.6)
31.5
52.9
15.6
21.2
47.2
31.6
26.7
47.2
26.1
26.1
52.7
21.2
30.7
37.3
31.9
23.9
56.7
19.3
<0.001
Anxiety 401 (12.7) 10.3 15.3 12.1 13.3 14.5 11.8 0.031
Depression 629 (19.9) 25.5 14.2 19.9 20.0 19.3 20.4 0.482
In the same way, some kind of analysis of chronic conditions would be interesting.
R: Below is added
Future studies should also investigate the relationship between sedentary behaviour and physical activity with anxiety and depression in patients with specific chronic conditions.
Reviewer 3 Report
The revisions made by the authors are satisfactory and I recommend this brief report to be published in IJERPH after through copyediting.
Author Response
The revisions made by the authors are satisfactory and I recommend this brief report to be published in IJERPH after through copyediting.
R: Corrected